# Pharmacokinetics of Shikimic Acid Following Intragastric and Intravenous Administrations in Rats

**DOI:** 10.3390/pharmaceutics12090824

**Published:** 2020-08-29

**Authors:** Keumhan Noh, Hyun-Moon Back, Beom Soo Shin, Wonku Kang

**Affiliations:** 1Deapartment of Pharmaceutical Sciences, Leslie Dan Faculty of Pharmacy, University of Toronto, Toronto, ON M55 3M2, Canada; keumhan.noh@toronto.ac.ca; 2Department of Pharmaceutics, Ernest Mario School of Pharmacy, Rutgers, The State University of New Jersey, Piscataway, NJ 08854, USA; hyunmoon.back@rutgers.edu; 3School of Pharmacy, Sungkyunkwan University, Suwon 16419, Korea; 4College of Pharmacy, Chung-Ang University, Seoul 06974, Korea

**Keywords:** shikimic acid, HPLC-MS/MS, intragastric, intravenous, pharmacokinetics

## Abstract

Shikimic acid, a critical starting material for the semi-total synthesis of oseltamivir to treat and prevent influenza, exerts many pharmacological effects. However, the optimal bioanalytical method has not been adequately defined. We used liquid chromatography-tandem mass spectrometry to quantitate shikimic acid in rat plasma and studied its pharmacokinetics after intragastric and intravenous administration. Plasma was spiked with an internal standard, and the proteins were precipitated with acetonitrile, followed by solvent evaporation and reconstitution of the mobile phase. Shikimic acid was separated on a hydrophilic reverse-phase column and showed a mass transition ([M-H]^−^) at m/z 173.4→136.6. Shikimic acid exhibited bi-exponential decay after intravenous dosing, with a rapid distribution (5.57 h^−1^) up to 1 h followed by slow elimination (0.78 h^−1^). The steady state distribution and clearance volumes were 5.17 and 1.79 L/h/kg, respectively. After intragastric administration, the shikimic acid level peaked at about 3 h, and the material then disappeared mono-exponentially with a half-life of 1.3 h. A double peak phenomenon was observed. The absolute oral bioavailability was about 10% in rats. We explored the relationship between the pharmacokinetics and pharmacodynamics of shikimic acid.

## 1. Introduction

Shikimic acid (3,4,5-trihydroxy-1-cyclohexene-1-carboxylic acid) was initially extracted from *Illicium* species, including *I. anisatum* and *I. verum* [1,2]. Shikimic acid is a critical starting material for semi-total synthesis of oseltamivir, which is used to treat and prevent influenza caused by the A and B viruses [1].

Shikimic acid also exhibits anti-microbial, anti-inflammatory, and analgesic actions and is a major constituent of plant stem cells [3,4]. Shikimic acid prevents nerve demyelination by promoting the differentiation of oligodendrocyte precursor cells [5] and increases the levels of cytokines that promote hair growth [6]. However, the pharmacokinetic behavior of shikimic acid has not been fully elucidated.

The absorption of shikimic acid has only been studied via in situ perfusion in rats [7], while the metabolism was explored over 40 years ago [8,9]. Adamson et al. reported that shikimic acid was converted into hippuric acid (via benzoate) after aromatization by the gut microflora of rats and monkeys [8]. However, this was disputed by Brewster et al. several years later, who showed that shikimate was biotransformed by the gastrointestinal microflora into cyclohexanecarboxylate, which was then aromatized and conjugated with glycine in mammalian tissues [9].

Pharmacokinetic studies require sensitive analytical methods. Shikimic acid levels in several berries [10] and spring wheat [11] have been quantified by high-performance liquid chromatography (HPLC) with ultraviolet light detection, while the levels in plant extracts were evaluated by orbitrap mass spectrometry [12], LC, and gas chromatography (GC) coupled with isotope dilution mass spectrometry (MS), or GC-MS [13,14]. Recently, LC-MS/MS has been used to describe the chemical constituents of *Smilax glabra* Roxburgh and *Smilax china* Linn [15]. However, no method for analyzing shikimic acid in plasma has been reported.

Therefore, we developed and validated a quantitative analytical method for determining shikimic acid levels in rat plasma via HPLC-MS/MS, assessed shikimic acid stability during the pre-treatment and storage of plasma samples, and performed a pharmacokinetic study following intragastric (ig) and intravenous (iv) administration of shikimic acid to rats.

## 2. Materials and Methods

### 2.1. Materials

Shikimic acid, diclofenac and formic acid were purchased from Sigma-Aldrich (St. Louis, MO, USA), and acetonitrile was obtained from Burdick & Jackson (Muskegon, MI, USA). All other chemicals and solvents were of the highest analytical grade available.

### 2.2. Quantification of Shikimic Acid in Plasma

Standard solutions of shikimic acid and diclofenac (the internal standard; IS) were prepared in distilled water and methanol, respectively (1 mg/mL), and serially diluted. Then, 10 ng/mL of each solution was infused into the mass spectrometer at 10 μL/min and the major product ions were identified. Precursor ions and fragmentation patterns were monitored in negative ion mode. The major mass spectrometry peaks were used to quantify the compounds. The shikimic acid standard solution was serially diluted in distilled water to yield solutions of 50 ng/mL to 50 μg/mL. Then, 10 μL of each sample was added to 90 μL of drug-free plasma (final concentrations of 5, 20, 50, 100, 500, 1000, and 5000 ng/mL). Acetonitrile (0.5 mL) and 100 ng/mL diclofenac were used to spike the plasma samples. The mixtures were stirred vigorously for 10 s and then centrifuged at 12,000× *g* for 10 min at 4 °C. Each supernatant was transferred to a tube, and the liquid was evaporated under reduced pressure at 50 °C. The residue was reconstituted with 50 μL of the mobile phase. Each solution was filtered and 5 μL was injected into the LC-MS/MS system. Linear regression was used to derive five calibration graphs based on the ratio of the areas under the peaks for shikimic acid and the IS.

An API 4000 LC-MS/MS system (AB SCIEX, Framingham, MA, USA) with an electrospray ionization interface was used to quantify the IS and shikimic acid. The substances were separated on a reverse-phase column (SeQuant ZIC-HILIC; 150 × 2.1 mm internal diameter, 5 μm particle size; Merck KGaA, Darmstadt, Germany) at 30 °C. The mobile phase was a mixture of distilled water and acetonitrile (3:7 *v/v* with 0.1% formic acid). The elution rate was 0.2 mL/min and was controlled by an HP 1260 series pump (Agilent, Wilmington, DE, USA); the total run time was 5 min. The turbo ion spray interface was maintained at 4500 V and 450 °C. Multiple reaction monitoring (MRM) was used to quantify the precursor and its product ions, and the ratio of the peak areas for each solution was calculated. All data were processed by Analyst software (ver. 1.5.2; Applied Biosystems, Foster City, CA, USA).

### 2.3. Method Validation and the Stability of Shikimic Acid

Drug-free plasma from five rats was used to assess specificity. Shikimic acid quality controls were prepared at 4 different concentrations (5, 30, 500, and 3000 ng/mL for lower limit of quantification, low, intermediate and high concentrations, respectively). The precision and accuracy of intra- and inter-day assays were evaluated at these concentrations. Acceptable ranges were ±20% for the lower limit of quantification and ±15% for the other quality controls. Recovery was calculated by dividing the peak areas of quality controls of pre-spike by those of post-spike. A possible matrix effect was examined by dividing the peak areas of post-spike by those in neat solutions [16].

The stability of shikimic acid at concentrations of 0.1 and 1 μg/mL in rat plasma was examined under various conditions. The plasma samples were stored at room temperature for 4 h and at −70 °C for 3 weeks to assess short- and long-term stability, respectively. Stability was assessed after three freeze-thaw cycles (from −70 °C to room temperature) and in extracts stored for 24 h at 4 °C. The stability values were derived by comparing the peak ratios of stored and freshly prepared samples. Variation of ±15% was considered acceptable.

### 2.4. Animal Study

We used 10 male, 9-week-old Sprague Dawley rats weighing 250–260 g. The animal room was maintained at 23 °C, with a relative humidity of 50 ± 10%, 10–20 air changes/h, and a light intensity of 300 Lux under a 12 h/12 h light-dark cycle. The study was approved by the Institutional Animal Care and Use Committee (IACUC) of Chung-Ang University (No. 202000054, 20, 05, 2020). All animals were cared for in accordance with the principles of the National Institutes of Health Guide for the Care and Use of Laboratory Animals. The 10 rats were randomly assigned to the ig and iv groups (*n* = 5/group). Shikimic acid was dissolved at 10 mg/mL in saline for ig administration (100 mg/kg, 1 mL/kg) and at 2 mg/mL in saline for iv administration (2 mg/kg, 1 mL/kg). To measure plasma concentrations, blood samples (250 µL) were collected from the subclavian veins at 2, 5, 15, and 30 min and 1, 1.5, 2, 3, 4, 6, and 8 h after iv administration, and at 0.25, 0.5, 0.75, 1, 1.5, 2, 3, 4, 6, and 8 h after ig administration. The samples were then heparinized, centrifuged at 17,000 rpm for 10 min, and stored at −70 °C prior to analysis. Plasma was pre-treated as described above, and the amount of shikimic acid in the 100 µL samples was determined.

### 2.5. Pharmacokinetic Data Analysis

The pharmacokinetic parameters of shikimic acid were calculated from the time courses of its plasma concentrations. The maximum concentration (C_max_) and time to C_max_ (T_max_) were read from the individual data. The elimination rate constant (k) was estimated by linear regression of the log-transformed plasma shikimic acid concentration in the terminal phase, and the trapezoidal rule was used to obtain the area under the plasma concentration-time curve (AUC_t_). The AUC_inf_ was calculated by adding C_last_/k to AUC_t_, and clearance was given as the dose/AUC_inf_.

The apparent initial distribution rate constant (α) was estimated by log-linear regression of the residuals between the actual plasma concentration during distribution and the extrapolated plasma concentrations from the terminal phases, and the intercept (A) of the curve was used to predict the initial distribution volume. The volume of distribution (V_ss_) in the steady state was given by the dose divided by the sum of A and the intercept (B) of the line of the log-transformed concentration in the terminal phase. All data are shown as mean and standard deviation.

## 3. Results and Discussion

### 3.1. Validation of the Bioanalytical Method for Determining Shikimic Acid Levels in Rat Plasma

Shikimic acid is hydrophilic; the water solubility is 18 g/100 mL at 23 °C, so the substance is freely soluble (between very soluble and soluble) in water [17]. However, shikimic acid is practically insoluble in highly lipophilic organic solvents such as petroleum ether, chloroform, and benzene. Therefore, we initially ruled out the use of a liquid-liquid extraction method for purification of the substance from plasma. Instead, we tested solid-phase extraction using an anion exchange column, but the recovery rate was very low. We tested a simple precipitation procedure using a mixture of methanol and 10% ZnSO4 aqueous solution (4:1, *v*/*v*) [18], trichloroacetic acid (5% *w*/*v*) [19], methanol, and acetonitrile separately [20,21,22], but the sensitivity was poor. Therefore, the organic layer was evaporated after protein precipitation with five volumes of acetonitrile, and the plasma samples were finally concentrated twofold.

Figure 1 shows the chemical structure of shikimic acid and its product ion mass spectrum. The precursor ion of shikimic acid in negative mode was superior to that in positive mode. Deprotonated shikimic acid ([M-H]^−^ at *m*/*z* 173.4) was fragmented into its largest product ion at *m*/*z* 136.6, with a collision energy of −16 eV. The transition from the IS precursor to the [M-H]^−^ product ion occurred at 296.1→251.7 (collision energy −18 eV), as reported previously [20,21,22].

An isotope of the analyte would be an ideal IS. ^14^C- or ^13^C-labeled shikimic acid was used in a few previous studies [9,13]. However, no isotope of shikimic acid is currently commercially available. Although a synthetic method labeling shikimic acid with deuterium at positions C3 and C4 is available [23], it is impractical, as there are many steps and a low yield. Therefore, diclofenac served as a stable IS, as in many previous reports from our laboratory [20,21,22].

Typical chromatograms of shikimic acid and the IS in rat plasma are shown in Figure 2. No significant endogenous interference was evident at the elution time of either substance (Figure 2A). Shikimic acid and the IS eluted at 2.63 and 1.77 min, respectively (Figure 2B,C). On conventional reverse-phase columns (e.g., C_18_ and C_8_), shikimic acid eluted before 1 min, and sensitivity was compromised by ion suppression caused by endogenous substances eluting at around the same time. Therefore, we used a hydrophilic stationary phase for retention and adequate sensitivity. Shikimic acid was more hydrophilic than the IS, and it was retained for longer by the stationary phase.

The shikimic acid-IS peak area ratios exhibited a strong linear relationship with the corresponding plasma concentrations from 5 ng/mL to 5 μg/mL (*y* = (0.149 ± 0.030) x + 0.005 ± 0.002, r^2^ > 0.998). The detection limit was 2 ng/mL at a signal-to-noise ratio of 3. The intra- and inter-day assay accuracy and precision were within acceptable ranges (Table 1).

The extraction recovery was 84 ± 5%, and any matrix effect caused by endogenous plasma materials was negligible (92 ± 9%). Our assay reliably determined the shikimic acid concentration in rat plasma. Shikimic acid was stable in plasma under the tested conditions (Table 2). Although some instability was evident after long-term storage, it was within the acceptable range.

### 3.2. Time Courses of Plasma Shikimic Acid Concentrations Following Iv and Ig Administration

The mean time courses of plasma shikimic acid concentrations after iv and ig administration are shown in Figure 3 and Figure 4, respectively. The pharmacokinetic parameters are listed in Table 3.

Shikimic acid exhibited bi-exponential decay after iv administration, with a fast distribution (5.57 h^−1^) up to 1 h followed by slow elimination (0.78 h^−1^). The mean initial and steady state distribution volumes were 0.51 and 5.17 L/kg, respectively. The mean clearance was 1.79 L/h/kg. After ig administration, the plasma shikimic acid level peaked at 904 ± 48 ng/mL after about 3 h (2.7 ± 1.5 h) and then declined mono-exponentially (half-life = 1.3 ± 0.2 h). Absorption seems to be relatively slow, given the mean T_max_. However, as shown in the inset of Figure 4, three of five rats exhibited rapid absorption; the first peak appeared within 1 h and the second peak tended to be higher than the first.

As shown in Figure 4, the interindividual variation during absorption was much greater than in the other phases, attributable to a significant double peak phenomenon (perhaps reflecting enterohepatic circulation and/or belated absorption by different regions of the gastrointestinal tract). In contrast to the terminal phase after iv administration, we did not observe bi-exponential decay, not only because the distribution was relatively short (and its phase thus insignificant), but also because absorption persisted for up to 3 h. The AUC_inf_ was over 99% of the AUC_t_, indicating that the course of plasma sampling was sufficiently long to reveal the full time course of the plasma shikimic acid concentration. The total clearance of 1.73 ± 0.1 L/h/kg was similar to that after iv administration. The absolute oral bioavailability of shikimic acid was about 10%.

A couple of reports on the pharmacological action of shikimic acid in vivo have appeared [14,24,25]. Shikimic acid was administered orally to rats with streptozotocin-induced diabetes at 50 and 100 mg/kg [14], and 30 and 50 mg/kg [24]. It showed a promising anti-diabetic action comparable to those of metformin and glipizide and prevented retinopathy by exerting an antioxidant effect. At 25 and 50 mg/kg, shikimic acid also reduced the severity of cerebral ischemic injury induced by arterial thrombosis [25]. As the extent of systemic exposure to shikimic acid remained unclear, we evaluated a pharmacological dose (100 mg/kg) to elucidate the relationship between systemic exposure and pharmacodynamics.

## 4. Conclusions

We devised a new bioanalytical method for determining the shikimic acid level in rat plasma using LC-MS/MS and conducted a pharmacokinetic study involving both iv and ig administration. The absolute bioavailability was about 10%. Our findings will facilitate further work on the relationship between the pharmacokinetics and pharmacodynamics of shikimic acid.

## Figures and Tables

**Figure 1 pharmaceutics-12-00824-f001:**
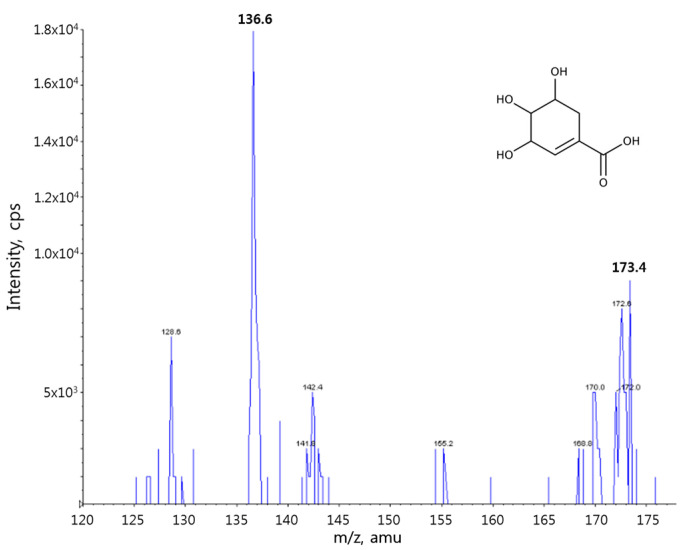
Chemical structure and fragment mass spectrum of shikimic acid.

**Figure 2 pharmaceutics-12-00824-f002:**
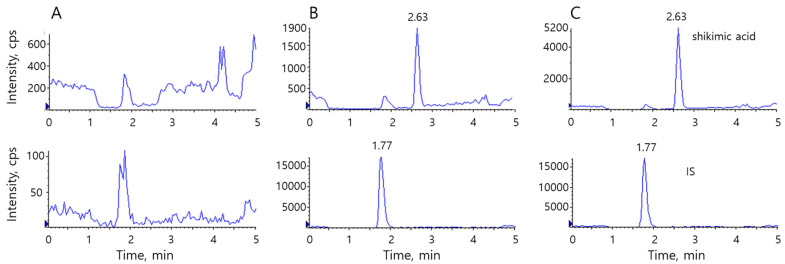
Typical chromatograms of shikimic acid (top) and diclofenac (internal standard (IS), bottom). (**A**) double blank, (**B**) spiked plasma with 10 ng/mL of shikimic acid, (**C**) a real sample, calculated to be 23 ng/mL.

**Figure 3 pharmaceutics-12-00824-f003:**
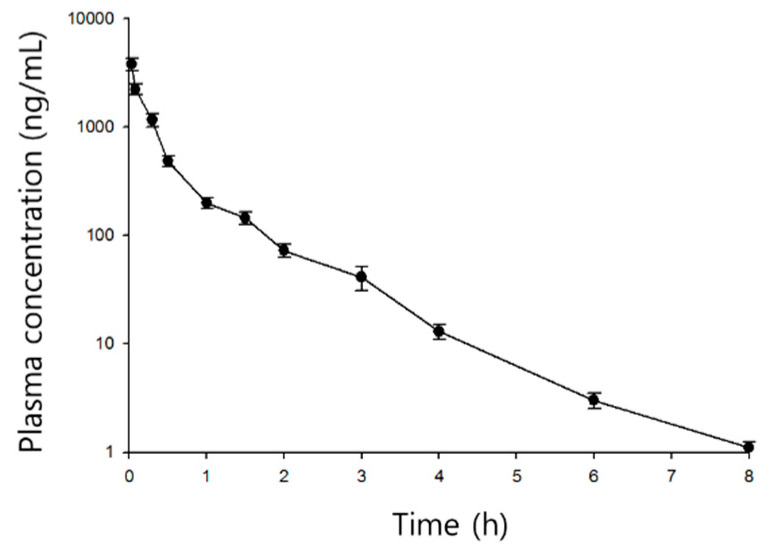
Time course of plasma shikimic acid concentrations following intravenous (2 mg/kg) administrations in rats (mean ± S.D., *n* = 5).

**Figure 4 pharmaceutics-12-00824-f004:**
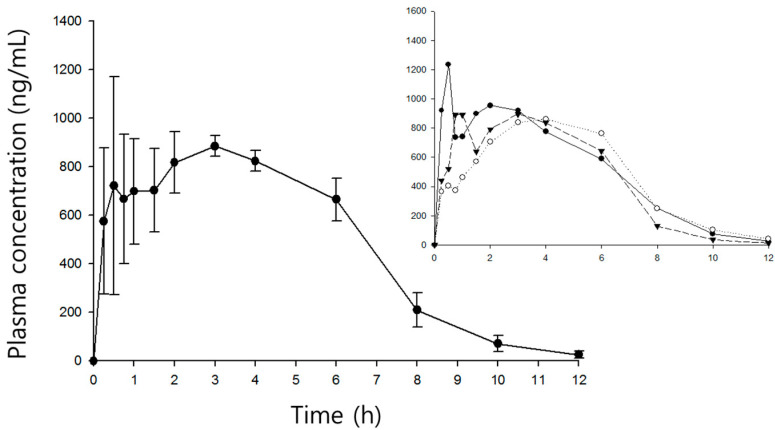
Time course of plasma shikimic acid concentrations following ig (100 mg/kg) administrations in rats (mean ± S.D., *n* = 5). Insert depicts three individual profiles significantly representing double peak phenomena.

**Table 1 pharmaceutics-12-00824-t001:** Assay validation of shikimic acid in rat plasma.

Concentration (ng/mL)	Intra-Day	Inter-Day
5	98.3 ± 5.6 ^a^ (5.7) ^b^	97.4 ± 6.8 (7.0)
30	102.2 ± 4.5 (4.4)	98.6 ± 7.2 (7.3)
500	103.5 ± 6.3 (6.1)	102.5 ± 5.7 (5.6)
3000	99.5 ± 6.4 (6.4)	97.4 ± 5.6 (5.7)

^a^ accuracy (mean % ± S.D., *n* = 5); ^b^ relative standard deviation (%).

**Table 2 pharmaceutics-12-00824-t002:** Stability of shikimic acid in rat plasma under various conditions.

Conditions for Stability Test	100 ng/mL	1000 ng/mL
room temperature for 4 h	95.6 ± 4.5 ^a^	99.5 ± 5.2
3-cycle freeze-thaw	106.5 ± 7.6	98.6 ± 6.2
post-extraction at 4 °C for 24 h	98.5 ± 5.4	95.3 ± 5.3
−70 °C for 3 weeks	110.5 ± 9.2	108.2 ± 8.3

^a^ accuracy in plasma under various conditions (mean % ± S.D., *n* = 3).

**Table 3 pharmaceutics-12-00824-t003:** Pharmacokinetic parameter of shikimic acid after ig (100 mg/kg) and iv (2 mg/kg) administrations in rats (mean ± S.D., *n* = 5).

Parameter	Intra-Gastric (100 mg/kg)	Intravenous (2 mg/kg)
C_max_ (ng/mL)	904 ± 48	-
T_max_ (h)	2.7 ± 1.5	-
t_1/2_ (h)	1.26 ± 0.20	1.12 ± 0.15
AUC_t_ (mg·h/L)	5.76 ± 0.32	1.12 ± 0.06
AUC_inf_ (mg·h/L)	5.81 ± 0.32	1.13 ± 0.06
Cl (L/h/kg)	1.73 ± 0.10	1.79 ± 0.09
Vi (L/kg)	-	0.51 ± 0.06
Vss (L/kg)	-	5.17 ± 0.67
A (ng/mL)	-	3500 ± 385
α (h^−1^)	-	5.57 ± 0.61
B (ng/mL)	-	387 ± 50
β (h^−1^)	-	0.78 ± 0.10
bioavailability, F (%)	10.4	-

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
