# Peer review of "Pharmacokinetics of Shikimic Acid Following Intragastric and Intravenous Administrations in Rats"

_pharmaceutics, 2020, doi:10.3390/pharmaceutics12090824_

Round 1
Reviewer 1 Report
In my opinion is a very interesting work, I have no comments.
Author Response
There was no comment.
Reviewer 2 Report
The current manuscript describes the development of a new LC/MSMS methodology for the determination of shikimic acid after its intragastric and intravenous administration. The subject is interesting whereas the experiments are well designed and executed and the statistical treatment is adequate. nevertheless there are some points that should be taken into account
line 144 please specify the meaning of superior
line 165 please provide the calibration equation along with the related std errors of the slope and the intercept. Is the slope statistically different from 0? Did the authors use some kind of weighting (as the dynamic range is extensive?)
line 166 please specify the methodology used for the estimation of the noise
line 172 please specify the levels used for the extraction recovery and the matrix effect along with the related uncertainty
Author Response
line 144 please specify the meaning of superior
- It means that the intensity of the precursor ion was checked in both negative and positive modes as conducted normally. The intensity in negative mode was bigger than that in positive mode.
line 165 please provide the calibration equation along with the related std errors of the slope and the intercept. Is the slope statistically different from 0? Did the authors use some kind of weighting (as the dynamic range is extensive?)
- Mean equation of calibration curves with deviation is inserted: “The shikimic acid:IS peak area ratios exhibited a strong linear relationship with the corresponding plasma concentrations from 5 ng/mL to 5 μg/mL [y = (0.149±0.030) x + 0.005 ± 0.002, r2 > 0.998].” The slope was statistically significant, and no weighting factor was used. Therefore, no need to insert additional information.
line 166 please specify the methodology used for the estimation of the noise
- As mention in the text, peaks at the elution time of shikimic acid which is less than 3-fold of the detection limit peak were regarded as noise.
line 172 please specify the levels used for the extraction recovery and the matrix effect along with the related uncertainty
- Their means and standard deviations were added as follows: The extraction recovery was 84±5% and any matrix effect caused by endogenous plasma materials was negligible (92±9%).
Reviewer 3 Report
Article: Pharmacokinetics of shikimic acid following intragastric and intravenous administrations in Rats
Line 33-35: “Shikimic acid is a critical starting material for semi-total synthesis of oseltamivir, which is used to treat and prevent influenza caused by the A and B viruses”. It is not clear, what is the relationship between the shikimic acid and the oseltamivir to give this drug as example. What is the metabolism of oseltamivir after administration in the human body?
Line 52: “Rhizoma Smilacis Glabrae and Rhizoma Smilacis Chinae”; I suggest to give the scientific names of the two plants (genus and species names) and to write Rhizome instead the Latin name “rhizoma”.
Line 63: “Measurement of shikimic acid levels in plasma, it is not clear how can you make this measurement in plasma that is not pure solvent, it contains many constituents.
Line 94: “Recovery was estimated by comparing the peak areas of quality controls spiked before and after pretreatment”. This need figures to show this comparison.
Line 143: “Figure 1 shows the chemical structure of shikimic acid and its product ion mass spectrum”. Is this figure for test in blood plasma or in control test without plasma?
Author Response
Line 33-35: “Shikimic acid is a critical starting material for semi-total synthesis of oseltamivir, which is used to treat and prevent influenza caused by the A and B viruses”. It is not clear, what is the relationship between the shikimic acid and the oseltamivir to give this drug as example. What is the metabolism of oseltamivir after administration in the human body?
- Shikimic acid is not a metabolite of oseltamivir; the former is a starting material for the latter in the synthetic process.
Line 52: “Rhizoma Smilacis Glabrae and Rhizoma Smilacis Chinae”; I suggest to give the scientific names of the two plants (genus and species names) and to write Rhizome instead the Latin name “rhizoma”.
- Gizoma Smilacis Glabrae and Phizoma Smilacis Chinae were changed to Smilacis Glabrae Rhizome and Smilacis chinae rhizome, respectively: “Recently, LC-MS/MS has been used to describe the chemical constituents of Smilacis Glabrae Rhizome and Smilacis Chinae Rhizome [15].”
Line 63: “Measurement of shikimic acid levels in plasma, it is not clear how can you make this measurement in plasma that is not pure solvent, it contains many constituents.
- The title of Section 2.2 was changed to “2.2. Quantitation of shikimic acid in plasma”
Line 94: “Recovery was estimated by comparing the peak areas of quality controls spiked before and after pretreatment”. This need figures to show this comparison.
- The recovery as expressed in percentage should be enough.
Line 143: “Figure 1 shows the chemical structure of shikimic acid and its product ion mass spectrum”. Is this figure for test in blood plasma or in control test without plasma?
- The product ion spectrum is monitored in neat solution.
Round 2
Reviewer 3 Report
Article: (Pharmacokinetics of shikimic acid following intragastric and intravenous administrations in Rats)
Line 33-35: your answer about the first review included “Shikimic acid is not a metabolite of oseltamivir; the former is a starting material for the latter in the synthetic process”. This confirms our previous comment there is no relationship between oseltamivir and your study aim about the pharmacokinetic of shikimic acid. The synthetic process has no relationship with the metabolism of a dug in the human or animal body. Does oseltamivir release shikimic acid in the human blood? If yes, write this as paragraph with reference. If not; give other example for a drug which it is metabolized to or releases shikimic acid in the blood. Alternatively, you can speak about only shikimic acid as a drug that still unchanged in the blood.
Line 52: The scientific botanical names of the plants species are Smilax glabra Roxb. ; Smilax china L. (Smilacaceae), check this in: http://www.theplantlist.org
The names of the crude drugs obtained from these plants are “Rhizoma Smilacis Glabrae” and “Rhizoma Smilacis Chinae” according to the Chinese pharmacopoeia. However, may be these are not known in the international pharmacopoeias. Therefore I suggested it is better to write the scientific botanical names of the plant species, but in the "pharmaceutics" you can use the both names.
Line 65: “Quantitation of shikimic acid in plasma”; did you mean “Quantification” or “quantitative analysis” ?.
Author Response
Line 33-35: your answer about the first review included “Shikimic acid is not a metabolite of oseltamivir; the former is a starting material for the latter in the synthetic process”. This confirms our previous comment there is no relationship between oseltamivir and your study aim about the pharmacokinetic of shikimic acid. The synthetic process has no relationship with the metabolism of a dug in the human or animal body. Does oseltamivir release shikimic acid in the human blood? If yes, write this as paragraph with reference. If not; give other example for a drug which it is metabolized to or releases shikimic acid in the blood. Alternatively, you can speak about only shikimic acid as a drug that still unchanged in the blood.
- Altough shikimic acid is used as a starting material for oseltamivir, two hydroxy groups at C4 and C5 in cyclohexene ring of the former are substituted to amide and amine groups, respectively. Therefore, shikimic acid has not been found intact in human blood as a metabolite of osletamivir.
Line 52: The scientific botanical names of the plants species are Smilax glabra Roxb. ; Smilax china L. (Smilacaceae), check this in: http://www.theplantlist.org
The names of the crude drugs obtained from these plants are “Rhizoma Smilacis Glabrae” and “Rhizoma Smilacis Chinae” according to the Chinese pharmacopoeia. However, may be these are not known in the international pharmacopoeias. Therefore I suggested it is better to write the scientific botanical names of the plant species, but in the "pharmaceutics" you can use the both names.
- The scientific botanical names, Smilax glabra Roxburgh and Smilax china Linn were used on line 52.
Line 65: “Quantitation of shikimic acid in plasma”; did you mean “Quantification” or “quantitative analysis” ?.
- It has been changed to ‘Quantification’.